# Using Nanomaterials for SARS-CoV-2 Sensing via Electrochemical Techniques

**DOI:** 10.3390/mi14050933

**Published:** 2023-04-25

**Authors:** My-Van Tieu, Hien T. Ngoc Le, Sungbo Cho

**Affiliations:** 1Department of Electronic Engineering, Gachon University, Seongnam-si 13120, Republic of Korea; 2Department of Health Sciences and Technology, GAIHST, Gachon University, Incheon 21999, Republic of Korea

**Keywords:** SARS-CoV-2, electrochemical detection, biosensing platform, nanomaterials, virus detection

## Abstract

Advancing low-cost and user-friendly innovations to benefit public health is an important task of scientific and engineering research. According to the World Health Organization (WHO), electrochemical sensors are being developed for low-cost SARS-CoV-2 diagnosis, particularly in resource-limited settings. Nanostructures with sizes ranging from 10 nm to a few micrometers could deliver optimum electrochemical behavior (e.g., quick response, compact size, sensitivity and selectivity, and portability), providing an excellent alternative to the existing techniques. Therefore, nanostructures, such as metal, 1D, and 2D materials, have been successfully applied in in vitro and in vivo detection of a wide range of infectious diseases, particularly SARS-CoV-2. Electrochemical detection methods reduce the cost of electrodes, provide analytical ability to detect targets with a wide variety of nanomaterials, and are an essential strategy in biomarker sensing as they can rapidly, sensitively, and selectively detect SARS-CoV-2. The current studies in this area provide fundamental knowledge of electrochemical techniques for future applications.

## 1. Introduction

### 1.1. Introduction to Biosensors and Electrochemical Biosensors

The term “Biosensors” can have many definitions, but it generally comprises three parts: a part that can immobilize and analyze the target analytes, such as microorganisms, biomolecules (e.g., nucleic acids, antibodies, and hormones), or proteins; a transducer component that converts the physicochemical reaction to the flow of electrons; a display device (e.g., PC, laptop, or smartphone) for presentation of an output signal. Google Scholar database has indexed over 797,000 results with the keyword “biosensor”.

The first biosensor was invented by Cremer (1906) to measure the concentration of acid in solution by detecting the electric potential of the fluid situated on the opposite sides of a glass membrane [1]. In 1962, Clark Jr., famous for inventing the biosensor field, made an oxygen sensor in the blood on a platinum electrode protected by a piece of dialysis membrane [2]. Since then, incredible progress has been made. Guilbault and Montalvo developed a biosensor in a urea-specific enzyme electrode in 1969 [3], Yellow Springs Instrument Co., Inc. (YSI) (Yellow Springs, OH 45387, USA) launched the first commercially available glucose meter in 1972 [4], Liedberg and coworkers discovered a surface plasmon resonance (SPR) immunosensor for gas detection in 1983 [5], and I-Stat Corporation of Princeton, N.J. introduced six common blood tests with 90 s detection time in 1992 [6]. Considering the advances in the field of biosensors, this review aims to elucidate the electrochemistry and nanotechnology behind electrochemical sensors and their application in diagnosing severe acute respiratory syndrome coronavirus-2 (SARS-CoV-2) (Figure 1).

Until recently, electrochemical biosensing platforms have been the standard for point-of-care (POC) diagnosis of various human diseases because of their high sensitivity, specific measurements, ease of operation, and relatively rapid detection time ranging from tens of seconds to minutes. However, conventional methods, including quantitative reverse transcription–polymerase chain reaction (qRT-PCR), are ineffective in fulfilling the ideal conditions for sample detection because they rely on accurate cycle steps, expensive reagents, and are time-consuming (~3 h) methods (Figure 2). Therefore, electrochemical biosensors using electronics and nanotechnology have been developed (especially for the diagnosis of SARS-CoV-2), which can be commercialized as POC devices for clinical diagnostics. The final goal of developing electrochemical sensors is to reduce the sample volume needed for detection, varying from hundreds of nanoliters to several microliters, to conserve pretreatment sample reagents. To meet these requirements, electrochemical sensors have become a very promising scientific tool for research and development (Figure 3). Additionally, with this review, we hope to encourage further research in various infectious diseases employing nanotechnology and the electrochemical biosensor community.

### 1.2. Introduction to SARS-CoV-2

Common symptoms of SARS-CoV-2 are fever, cough, fatigue, and loss of sensation, while chronic cough, headaches, aches and pains, diarrhea, and eye irritation are less common. SARS-CoV-2 is caused by a coronavirus; it became a pandemic by late 2019, with more than 614,776,623 affected patients and about 6,536,187 deaths worldwide [8]. Coronavirus is an airborne pathogen but can also spread via hands exposed to infected surfaces that then touch the eyes, nose, or mouth. Scientists have tried to elucidate how SARS-CoV-2 causes COVID-19, but the lack of a viable treatment for COVID-19-associated acute respiratory distress syndrome illustrates that there is still much to be done [9]. WHO has suggested using Greek letters, such as alpha, beta, gamma, delta, and omicron to communicate about the emerging strains of this virus with non-scientific viewers. Table 1 lists the details of the reports on variants of SARS-CoV-2.

The respiratory tract samples for COVID-19 diagnosis are typically collected using nasopharyngeal swabs (from the back of the throat via the nose) and oropharyngeal swabs (from the middle of the throat (pharynx) slightly beyond the mouth). Saliva samples are obtained by spitting into a tube rather than using a nose or throat swab. Antibodies are only evaluated in the blood samples and are not used to diagnose COVID-19. The public can use the current diagnostic and therapeutic approaches, such as COVID-19 diagnostic tests, to collect their own samples at home using a collection kit following the Centers for Disease Control and Prevention guidelines and mail it to the medical laboratory for SARS-CoV-2 testing. Some test kits provide results in minutes at home without needing to send the samples to a laboratory. The current methods for SARS-CoV-2 diagnosis, the in vitro diagnostic (IVD) devices, are tests performed on samples collected from individuals. IVDs can be used to track diseases or other events as well as monitor a person’s overall health to heal, cure, or prevent illness [12]. SARS-CoV-2-related IVDs are classified into several types based on the method of coronavirus testing.

Molecular test: genetic material (RNA) from the virus can be amplified so that viral infection can be detected using PCR.Antibody test: SARS-CoV-2 infection is detected using Y-shaped molecules generated by the immune system to deactivate a virus and mark it for elimination from the blood.Antigen test: parts of a pathogen recognized by the immune system are called antigens; they can be detected within 15–30 min.

## 2. Electrochemical Biosensing Hotspots

In this issue, we highlight the laboratory’s contributions to the development of a wide range of electroanalytical techniques, such as cyclic voltammetry, electrochemical impedance spectroscopy, differential pulse voltammetry, and chronoamperometry for the successful detection of SARS-CoV-2.

### 2.1. Electrochemical Impedance Spectroscopy (EIS)

This technique works on a concept similar to resistance (the ability of a circuit to resist current). Resistance is a concept for ideal resistors. However, because many circuits are more complex, impedance has replaced resistance. Impedance considers all the elements of a perfect resistor, as well as inductance, resistance, and capacitance. EIS involves applying an alternating current (AC) source to a sample at various frequencies and measuring the electrical current. The ratio of the frequency-dependent potential (E) to the frequency-dependent current (I) is then used to compute the impedance (Z) (i.e., Z = E/I) [13]. Multiple frequency measurements are conducted using this method. EIS is non-destructive, extremely powerful, and sensitive and can be used to simultaneously investigate multiple electrochemical processes, e.g., the rate of transfer of electrons in a process, diffusion-limited reactions, or a system’s capacitive behavior at the electrode interface. Recent research has shown that EIS can be used to detect the corrosion of metals [14,15], food quality [16,17,18], and ion mobility [19]. The Faradaic impedance method has high precision and is commonly used for evaluating heterogeneous charge transport and for studying double-layer structures. Additionally, electrochemical biosensors based on EIS have various advantages, including low power consumption and ease of miniaturization. EIS-based biosensors produce signal output by utilizing periodic small AC perturbations. They respond to signal changes caused by the binding of bioanalytes to the immobilized biorecognition parts on the electrode surface. Because of these benefits, biosensors based on EIS can help develop miniaturized POC testing applications for medical devices. Salahandish et al. reported a bipotentiostat readout biosensor for SARS-CoV-2 nucleocapsid proteins in both spiked samples and clinical nasopharyngeal swab samples, with prospective uses in POC screening (Figure 4B) [20]. Two working screen-printed electrodes (SPEs) were fabricated using a graphene@PEDOT:PSS hybrid ink electroplated on the surface. The device performance was rapid (~30 min), giving repeatable and accurate measurements and performing comparably with or outperforming the commercial Autolab potentiostat. Tabrizi’s group used carbon nanofibers (CNFs) with embellished gold nanoparticles (AuNPs) on carbon SPE (CSPE) and measured EIS to detect SARS-CoV-2 receptor-binding domain with the limit of detection of 7.0 pM in the range of detection of 0.01–64 nM [21]. Table 2 summarizes the recent electrochemical impedance spectroscopy techniques used for the quantification of SARS-CoV-2.

### 2.2. Differential Pulse Voltammetry (DPV)

In this methodology, the current is measured before every potential change and the current change is shown as a function of potential. The background is rarely dominated by the charging current for solid electrodes of all types, but rather by Faradaic processes depending on the electrode material, solvent, or supporting electrolyte. By taking the difference between the current samples, DPV allows for the moderation of background contributions. However, the residual background is typically higher than that of mercury, sometimes decreasing the method’s sensitivity. Because its readout format allows the separation of signals from individual components along a common baseline, DPV is particularly well-suited to analyze multicomponent systems. DPV can be used with a much broader range of analytes than just heavy metal species. However, DPV requires a computer-controlled or programmable potentiostat [35].

DPV has two key advantages over traditional linear sweep voltammetry. First, it is far more sensitive. For each species being studied, the differential pulse approach produces a distinct peak and measuring the height or area of this peak and linking it to the concentration gives more reliable results than measuring the height of the current in the linear sweep mode. This is especially obvious at low concentrations, where a slight peak in the differential pulse mode is considerably easier to detect than a small shift in current height. The second advantage is that it is considerably easier to distinguish two species with comparable E1/2 values [36]. Recently, Nascimento et al. reported an ultrasensitive disposable electrochemical biosensor to profile the SARS-CoV-2 spike protein based on a magneto-assay [22]. As displayed in Figure 4A, the synthesis of magnetic beads (MBs) and gold nanoparticles (AuNPs) conjugated to ACE2 peptide via a straightforward thiol-Au bond allows for the capture and separation of spike proteins in saliva samples. Using transmission electron microscopy, the authors proved that AuNPs can be found on the surface of the MBs and showed biosensing ability with significantly low LOD of 0.35 ag/mL (60 min of detection time using DPV) that could work with saliva samples. Lima et al. also performed similar experiments investigating the effects of pencil graphite electrodes as low-cost electrochemically advanced diagnostic tools (~USD 1.50/unit) with a LOD viral spike protein of 229 fg/mL that could be correlated to the different concentrations of clinical samples [37]. Furthermore, DPV has been employed to study SARS-CoV-2 as a model electrochemical biosensor (Table 3).

### 2.3. Chronoamperometry (CA)

In chronoamperometry, the current is measured as a function of time, following a potential step perturbation when a squarewave potential is applied to the working electrode. Typically, the working electrode is stepped from a potential without any electrode reaction to one corresponding to the mass-transport-limited current and the ensuing current–time transient is measured. In double-step chronoamperometry, a second step inverts the electrode reaction, making it useful for assessing the circumstances in which the product of the original electrode reaction is consumed in solution by a related homogeneous chemical process. Nonetheless, the depletion effect remains, which means that the current continues to decay with time, as it does in any RC circuit, as described by the Cottrell equation. Notably, chronoamperometry has a higher signal-to-noise ratio than other amperometry techniques [48,49]. Table 4 contains information on other studies on using chronoamperometry to detect SARS-CoV-2. As illustrated in Figure 4C, they used a free reagent on a sensor-modified electrode chip, which was later modified with a DNA–antibody complex. The developed system is the first standalone sensor chip capable of detecting complete SARS-CoV-2 virus particles in pure saliva samples from COVID-19 infected individuals in minutes [23].

### 2.4. Cyclic Voltammetry (CV)

CV, as shown in the Figure 1, is an electrochemical technique that measures the current as a response to applied voltages and is a type of potentiodynamic electrochemical measurement. In general, the procedure is a reversible potential-controlled experiment that scans the electric potential before reversing directions after reaching the final potential and scans back to the initial potential at a predetermined period [55]. The electron transfer rate constants can be estimated by analyzing the variation in the peak position as a function of the scan rate. Using this technique, it is possible to investigate the surface using the characteristic potential of the electrodes [35]. Sukjee et al. described electrochemical detection of inactivated SARS-CoV-2 in an aquatic environment [56]. In this study, GO was modified with a molecularly imprinted polymer to achieve greater interaction with the target virus. The designed biosensor could detect virus samples at sub-fM concentrations and select samples using SPE electrodes (e.g., negative controls, H5N1 influenza A virus, and non-imprinted polymers). Nonetheless, the measurement of peak currents in CV does not contribute to the correction of the charging current because it is typically uncertain, such as the rate constant of a coupled homogeneous reaction or the concentration of the electrolyte [57]. This limitation complicates the interpretation of data.

## 3. The Importance of Nanomaterials for Developing Biosensors for SARS-CoV-2

Signal amplification and improving the properties of electrochemical sensors with state-of-the-art nanomaterials are important for ultrasensitive biosensors. The three most extensively studied structures—one-dimensional (1D) and two-dimensional (2D) graphene and metal nanoparticles—have already demonstrated novel sensing capabilities that can be used in SARS-CoV-2 electrochemical sensors. Owing to the extensive literature in this area, only the selected nanomaterials mentioned in this study will be reviewed below.

Nanoparticles (<10 nm): Gold nanoparticles (AuNPs) and nickel nanoparticles (NiNPs) have a vital role in the creation of improved electrochemical systems, providing several advantages, including smooth functionalization with biomaterials (relatively high biocompatibility), good consistency, high surface area per unit volume as the material size decreases, enhanced electron transfer, and catalysis of electrochemical reactions. They are capable of providing precise, simple, rapid, and affordable biosensors with improved LOD and therefore can be used to make sensors for SAR-CoV-2 in an electrochemical system [58,59]. Moreover, the AuNPs give the electrode stability and protection against harmful substances [60]. For example, Alafeef’s group fabricated a graphene-based paper electrochemical assay while using four thiol-modified antisense oligonucleotide (ssDNA) -capped AuNPs for the detection of nucleocapsid phosphoprotein (N-gene) of SARS-CoV-2 (less than 5 min) [50]. The importance of AuNP is reconfirmed once again in this study when comparing thiol-modified ssDNA-capped AuNPs on top of the gold electrode and without AuNP conjugation. The proposed assay showed a great sensitivity of 231 (copies μL^–1^)^−1^ and LOD = 6.9 copies/μL, demonstrated output stability and selectivity owing to its signal amplification system, and confirmed using the RT-PCR test using clinical samples (Figure 5). Another approach was reported by Zhao and coworkers; in 2021, Zhao et al. introduced a super sandwich-type electrochemical sensor for the detection of SARS-CoV-2 from infected COVID-19 patients using a smartphone. The sensing mechanism was on the basis of a sandwich assay in which two probes of capture and reporter were used to hybridize with the SARS-CoV-2 RNA and form a sandwich architect on an AuNPs@p-sulfocalix[8]arene functionalized graphene and AuNPs@magnetic nanocomposite on SPCE. The reporter probe was labeled with AuNPs@p-sulfocalix[8]arene functionalized graphene with toluidine blue to act as the redox reporter for electrochemical detection of SARS-CoV-2 RNA. The proposed sensor offered a great sensitivity of 3 aM, with a linear detection range from 10^−17^ to 10^−12^ M of the logarithmic SARS-CoV-2 RNA concentration and demonstrated great capability for clinical applications with 88 RNA samples [61]. Biosensors with magnetic nanoparticles have well-known capacities such as less background noise, good dispersions, and good biocompatibility relevant to the immobilized receptors [62,63]. Another approach was reported by Li and coworkers for a magnetic nanobead-based immunosensor coupled with a microfluidic device for SARS-CoV-2 nucleocapsid protein in serum [64]. For the electrode surface modification, the capture probe was immobilized on screen-printed gold electrode (SPGE) sensors. After the capture probe immobilization step, the dually labeled magnetic nanobeads were bound to the capture probe on SPGE sensors and conjugated with HRP-coated magnetic microbeads, which generated an amperometric current. The proposed device represented a rapid (<1 h) and handheld smartphone-based diagnostic device, with high sensitivity of 230 pg/mL in whole serum and 100 pg/mL in 5 × diluted serum and good selectivity, thanks to magnetic nanobeads, to serve as an immunomagnetic enrichment and signal amplification. Besides the nanoparticles mentioned above, many metal nanoparticles have been reported with the aim of developing sensitive electrochemical biosensors for SARS-CoV-2 detection [65], such as copper (Cu) and silver (Ag), etc.

Graphene and its derivatives: Graphene oxide (GO) and reduced graphene oxide (rGO) are well-known 2D carbon-based nanomaterials designed with a thick layer of organized single-atom carbon that have demonstrated their value in investigating the influence of electrode surface platforms on SARS-CoV-2 electrochemical sensors. The inherently good electrical conductivity and distinctive flat shape with a vast surface area boost the loading capacity of biomolecules. Furthermore, for experiments, graphene and its derivatives were modified with metal nanoparticles to improve the sensitivity and analytical performance of electrochemical biosensors [66,67,68]. However, the micro-sized structure elicits a strong inflammatory response in vitro and in vivo [69]. In a recent study, Zhang et al. developed a sensitive graphene field effect transistor (FET) for the detection protein of coronaviruses within 2 min. Their platform utilized a graphene structure along with highly sensitive antibody–antigen interaction to present very good sensitivity (LOD = 0.2 pM) and potential application in spike protein samples [70]. Li and coworkers discovered that graphene FET sensor with complementary phosphorodiamidate morpholino oligos modified on AuNP’s surface has also been employed to demonstrate the effective detection of SARS-CoV-2 RNAs samples. A relatively low LOD of 0.37 fM was achieved by the proposed biosensor with the ability to analyze SARS-CoV-2. Remarkably, this proposed sensor was given the advantage of detecting undiluted throat swabs and serum with high capability and precision in 2 min (Figure 6) [71]. Additionally, towards the fabrication of a graphene biosensing platform, Morawski’s group introduced a graphene-based 3D-printed six working electrode cell for point-of-care monitoring of three COVID-19 biomarkers by multiplex voltammetric [72]. In this approach, the graphene oxide was decorated onto 3D-printed six working electrode cells to exhibit very high sensitivity, down to 0.1 pg/mL, and performed in saliva and serum, due to the utilized graphene materials to improve the electrical conductivity and to support protein binding and the surface area of the electrode.

Carbon nanotubes: They are 1D nanomaterials and include multi-walled (MW) and single-walled (SW) carbon nanotubes that are appropriate for use in the construction of electrochemical sensors. Carbon nanotubes can provide strong electrocatalytic activity, strong chemical durability, high thermal stability, large specific surface area, improved electronic conductivity, and contamination resistance [72,73]. Moreover, carbon nanotubes provide direct electron transfer to bioreceptors, including antibodies and antigens [60]. Several studies have been carried out for detecting SARS-CoV-2. For instance, Shao et al. have demonstrated SW carbon nanotube-based FET for the detection of SARS-CoV-2 with a LOD of 0.55 fg/mL for spike antigen and 0.016 fg/mL for nucleocapsid antigen. As the sensor platform, the FET-modified SW carbon nanotube was modified by primary antibodies (anti-SARS-CoV-2 spike protein antibody and anti-nucleocapsid protein antibody) as the capture probe [74]. Cardoso and colleagues also conducted a study on detecting SARS-CoV-2 on modified CSPEs with SW carbon nanotubes. An SW carbon nanotube-based FET displayed an ultrasensitive detection signal to pave the way for the development of various forthcoming carbon nanotube-based immunosensors. As seen in Figure 7, carboxylated SW carbon nanotube electrode surface modifications were prepared. For the electrode surface modification, carboxylated SW carbon nanotubes were immobilized on SPCE by drying at 72 °C to serve as a highly conductive electrode. Carboxyl groups were activated by a solution of EDAC and NHS on SW carbon nanotubes to produce an amine layer capable of capturing the antibodies against SARS-CoV-2. The proposed platform was accumulated on the SPCEs, followed by a detectable electrochemical signal arising from potassium hexacyanoferrate III (K_3_[Fe(CN)_6_]) and potassium hexacyanoferrate II (K_4_[Fe(CN)_6_]) trihydrate. The proposed platform represented a simple and innovative biosensing platform with great sensitivity (LOD = 0.7 pg/mL) and selectivity, thanks to the diversified function of carbon nanotubes in signal enhancement (a simple and quick approach) to be a potential candidate for immunosensors [75].

These nanomaterials show that, just by varying their physical dimensions or outward methodologies such as the intensity and frequency of nanoelectrical sensory input, nanostructures can effectively interface with a wide range of bioreceptors, highlighting their potential use with various biotargets in the electrochemical sensor field. These sensing platforms functionalized with nanomaterials are planned to be employed as a fast screening tool in the emergency room. When close connections are being monitored by doctors, the approach can be utilized to swiftly identify diseased persons and provide doctors with timely feedback. This is thought to be a potent technique for controlling present COVID-19 infections, as well as any future outbreaks.

## 4. Concluding Remarks and Future Perspectives

In our previous work, we studied the clinical efficacy of biosensing technologies for the confirmation of SARS-CoV-2 infection [76]. Our group reviewed electrochemical biosensing technologies using nanomaterials that have shown excellent performance when applied to SAR-CoV-2. The first priority was to understand the principles and workings of electrochemical techniques. Established diagnostic tools for viral infections are limited by their long response times, exorbitant prices, lack of functionality, technical complexity, and poor sensitivity and specificity. Electrochemical biosensing technologies can detect their targets in small sample amounts and provide high detection accuracy at a cost comparable to the average income in less developed communities.

With advancements in ultrasensitive detection methods for electrochemical sensors, it may be possible to leverage these dynamic materials for functional POC biosensing applications in the future. The growing field of nanomaterial-based electrochemical biosensing technologies is rapidly expanding in terms of attention and practical implications for both research and clinical usage. Nanostructures can also address a broad range of interesting biological questions related to the impact of viral infections and biomolecules. Metal nanoparticles, carbon nanostructures, graphene, and their derivatives are still being studied for SARS-CoV-2-sensing applications. These endeavors will improve manufacturing and increase the applicability of nanomaterial-based electrochemical sensors. Consequently, nanostructures could be a powerful platform in biological research laboratories. Furthermore, biosensors based on nanomaterials can be designed to detect multiple targets simultaneously, allowing for more comprehensive and accurate diagnostics. With the ongoing research and development in this field, nanomaterial-enabled biosensors have the potential to become a powerful tool in the fight against COVID-19 and other infectious diseases.

Nonetheless, it should be acknowledged that emerging infectious diseases mainly prevalent in developing countries are generally overlooked by the research community. Governments, NGOs (non-governmental organizations), and private foundations all play important roles in attracting investment to mitigate these underappreciated infectious diseases. Furthermore, because their target markets are in less wealthy settings, commercialized POC devices endure challenges in obtaining funds from investors. We predict that mobile technology, in association with electronic diagnostic devices, will play a significant role in remote patient monitoring, particularly in less wealthy regions where healthcare is inaccessible. With all that in mind, electrochemistry, electrical impedance, and microfluidics will be extremely useful as a mobile phone readout methodology soon.

We believe that continued attempts by scientists and the biomedical industry to develop electrochemical biosensing technology will help millions of individuals who are disproportionately impacted by infectious diseases. We hope that this review will serve as a link between engineers and scientists interested in the use of nanotechnologies in biorelated applications.

## Figures and Tables

**Figure 1 micromachines-14-00933-f001:**
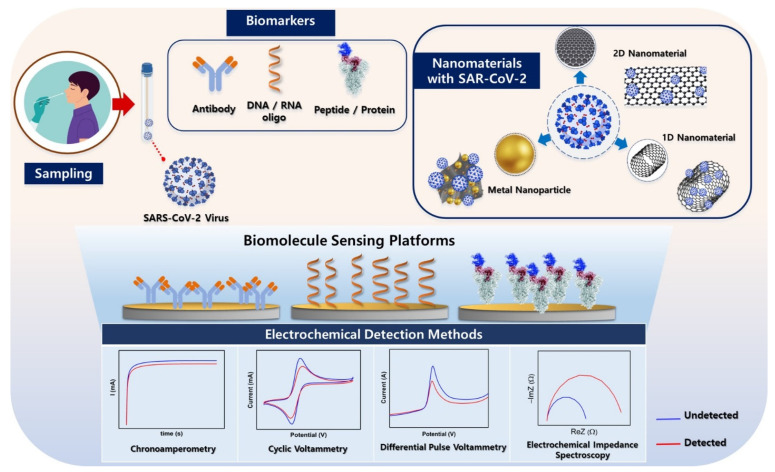
Schematic diagram representing the formation of nanomaterials with key targets and electrochemical detection methods to monitor successful platforms for the detection of SARS-CoV-2.

**Figure 2 micromachines-14-00933-f002:**
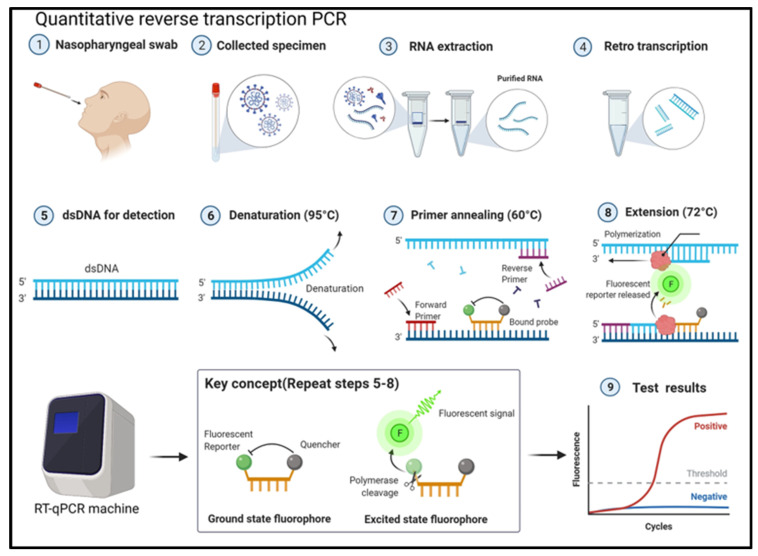
Schematic of quantitative reverse transcription–polymerase chain reaction (qRT-PCR) from the collecting sample step to the detection step. Reprinted with permission from Ref. [7]. Copyright 2022 Wiley Periodicals LLC.

**Figure 3 micromachines-14-00933-f003:**
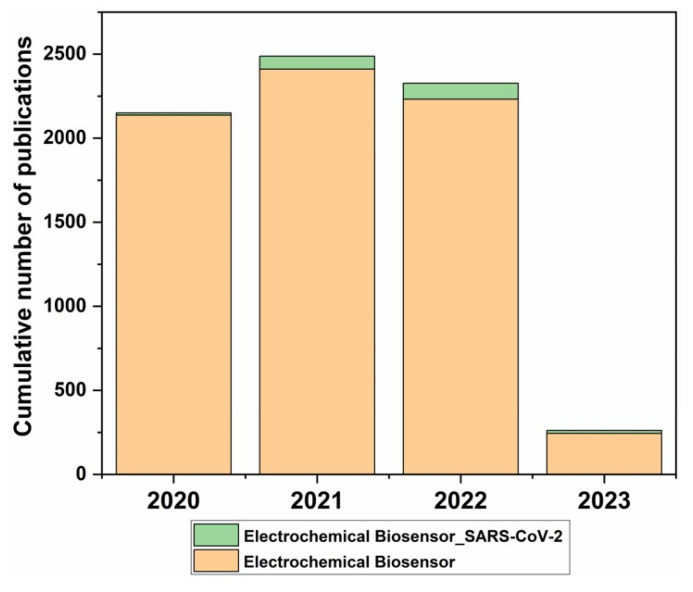
Cumulative publications on electrochemical biosensors and those capable of detecting SARS-CoV-2 from 2020 to 2023. The literature search was performed using the Web of Science. Keywords used: “electrochemical biosensor” and “electrochemical biosensor SARS-CoV-2”.

**Figure 4 micromachines-14-00933-f004:**
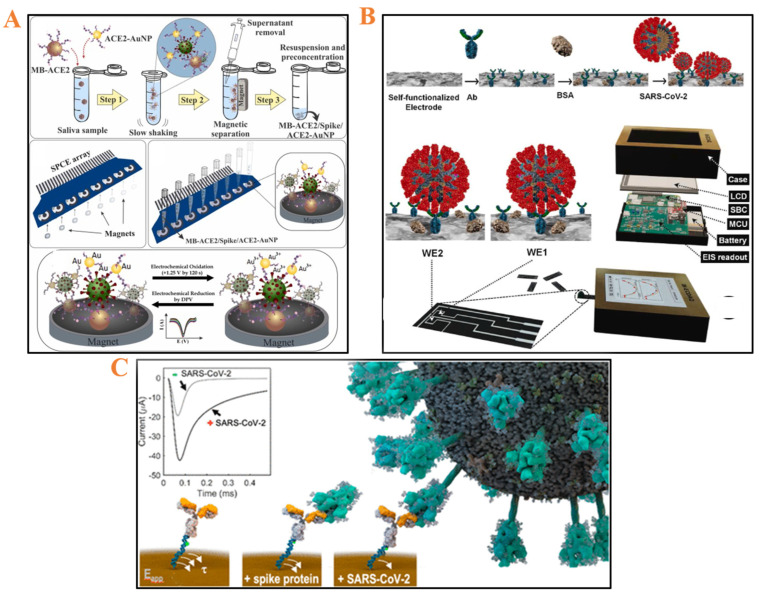
Electrochemical biosensors for detecting SARS-CoV-2. (**A**) A magneto−assay platform designed to discriminate SARS-CoV-2 using differential pulse voltammetry. Reprinted with permission from Ref. [22]. Copyright 2022 Elsevier Publishing Group. (**B**) Schematic of using electrochemical impedance spectroscopy with binary electrochemical data acquisition (Bi−ECDAQ). Reprinted with permission from Ref. [20]. Copyright 2022 Elsevier Publishing Group. (**C**) Chronoamperometry is used in reagent−free sensing of SARS-CoV-2 viral particles. Reprinted with permission from Ref. [23]. Copyright 2021 American Chemical Society Publishing Center.

**Figure 5 micromachines-14-00933-f005:**
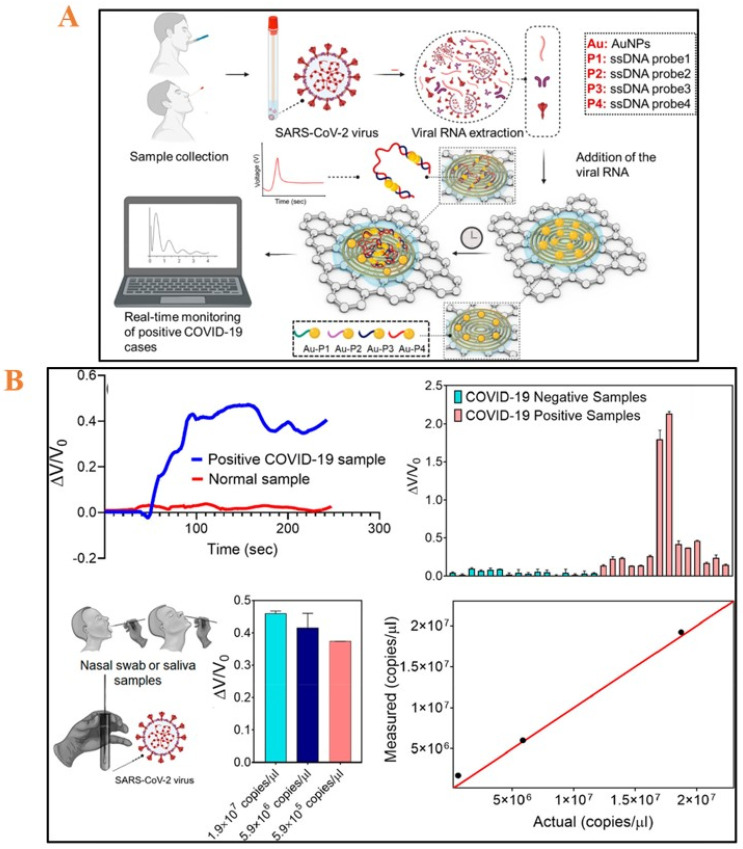
(**A**) Schematic diagram for screening of the detection of N-gene of SARS-CoV-2 graphene-based on electrochemical sensing platform. (**B**) The capability of the proposed electrochemical sensor to detect the presence of SARS-CoV-2 RNA in clinical samples. Reprinted with permission from Ref. [50]. Copyright 2020 American Chemical Society Publishing Center.

**Figure 6 micromachines-14-00933-f006:**
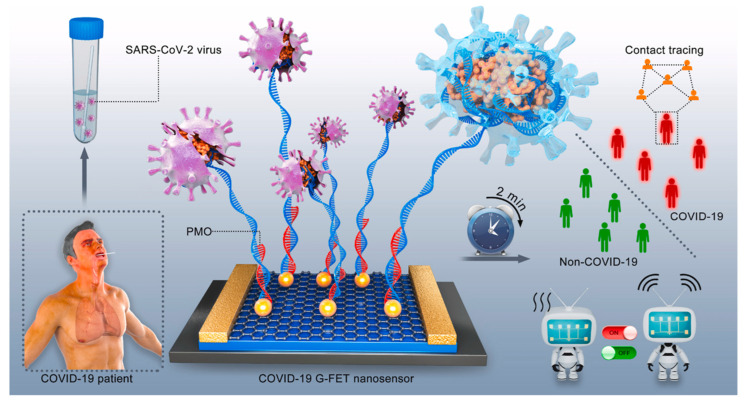
Schematic diagram for SARS-CoV-2 RNAs sample detection based on gold nanoparticle (AuNP)-decorated graphene field-effect transistor (FET). Reprinted with permission from Ref. [71]. Copyright 2023 Elsevier Publishing Group.

**Figure 7 micromachines-14-00933-f007:**
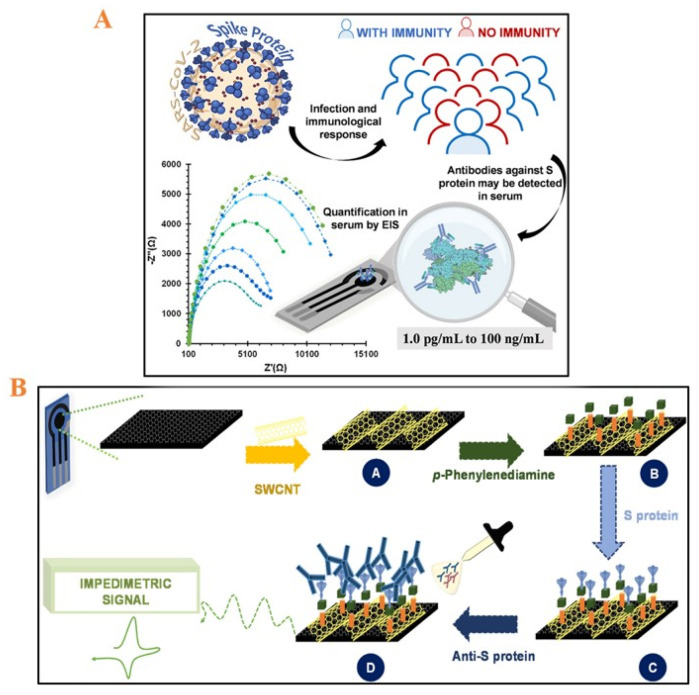
(**A**) Abstract of immunology assay for quantification of SARS-CoV-2 based on antibody-modified SW carbon nanotubes on CSPEs; (**B**) preparation of the proposed platform consists of SW carbon nanotubes delivering the protein captured *p*-phenylenediamine on SW carbon nanotubes on CSPEs to the detection including counter (carbon) and reference (silver) electrodes. Reprinted with permission from Ref. [75]. Copyright 2022 Elsevier Publishing Group.

**Table 1 micromachines-14-00933-t001:** Details of variants of SARS-CoV-2 [10,11].

Variant Name(s)	Earliest Documented Samples	Date of Designation	Severity	Vaccination Effectiveness
Omicron	Multiple countries	November 2021	Less severe disease than other variants	Yes
Delta	India	October 2020	More severe disease than other variants	Yes
Gamma	Brazil	November 2020	Hospitalization and death	Yes
Beta	South Africa	May 2020	Hospitalization and death	Yes (vaccines: AstraZeneca-Oxford, Pfizer-BioNTech, Moderna, and Johnson & Johnson also reported less protection against Beta)
Alpha	Great Britain	September 2020	Hospitalization, deadlier than the original virus	Yes (vaccines: Pfizer, Moderna, and Johnson & Johnson)

**Table 2 micromachines-14-00933-t002:** Representative examples of recent EIS for the detection of SARS-CoV-2.

SI. No.	Electrode Platform	Bioreceptors	Nanomaterials	Characteristics				Ref.
				LDR ^1^	LOD ^2^	Time Detection	Clinical Samples	
1	Screen-printed electrode (SPE)	Clinical samples	nr ^3^	1 pg/mL–100 ng/mL	2.8 fg/mL	4 min	Saliva and oropharyngeal/ nasopharyngeal swab	[24]
2	Indium tin oxide (ITO)	Antibody	AuNPs	0.0015 pg/mL–150 pg /mL	0.48 fg/mL	~90 min	Artificial nasal secretions	[25]
3	SPE	Antibody	Graphene@PEDOT:PSS	1 pg/mL– 10 ng/mL	116 fg/mL and 150 fg/mL	30 min	Nasopharyngeal sample	[20]
4	CSPE	Antibody	bbZnO/rGO	1–10,000 pg/mL	21 fg/mL	15 min	Nasopharyngeal swab	[26]
5	Au	Peptide	AuNPs	0.1–15 pM	0.1 pM	nr	Serum	[27]
6	Steel mesh electrodes	Antigen	AuNPs	1.0–2.5 × 10^3^ dilution factor	nr	30 min	Serum	[28]
7	Stainless steel mesh electrodes	Antigen	AuNPs	PPy:PSS/AuNPs: 10–60 ng/ mL and PPy-NTs/AuNPs: 0.4–8 ng/mL	PPy-NTs/AuNP: 0.386 ng/mL and PPy:PSS/ AuNPs: 2.456 ng/mL	nr	Serum	[29]
8	Interdigitated electrode	Aptamer	Carbon nanodiamond	1 fM–100 pM	0.389 fM	nr	Serum	[30]
9	SPE	Antibody	AuNPs	10–11–10^−7^ mol/ L	3.16 pmol/L (83.7 pg/mL)	35 min	Saliva	[31]
10	Au-SPE	Molecules	Molecularly imprinted polymer (MIP)	2.0–40.0 pg/mL	0.7 pg/mL	nr	Saliva	[19]
11	Carbon screen-printed electrode (CSPE)	Aptamer	CNF–AuNP	0.01–64 nM	7.0 pM	40 min	Saliva	[21]
12	Gold (Au) interdigitated electrodes	cpDNA	AuNPs	1.0 × 10^−18^–1.0 × 10^−6^ mol/L	0.5 aM~0.3 copy per mL	nr	nr	[32]
13	ITO	Antibody	AuNPs	0.002–100 pg/mL	0.577 fg/mL	nr	nr	[33]
14	Gold micropillar array electrodes	Antigen	rGO nanoflakes	0.01 fM–30 nM	SARS-CoV-2 spike S1 protein: 2.8 × 10–15 M and RBD: 16.9 × 10–15 M	nr	nr	[34]

^1^ Linear dynamic range (LDR). ^2^ Limit of detection (LOD). ^3^ nr—not reported.

**Table 3 micromachines-14-00933-t003:** Representative examples of recent DPV for the detection of SARS-CoV-2.

SI. No.	Electrode Platform	Bioreceptors	Nanomaterials	Characteristics				Ref.
				LDR ^1^	LOD ^2^	Time Detection	Clinical Samples	
1	Fluorine-doped tin oxide (FTO)	Antibody	AuNPs	1 fM–1 µM	0.63 fM	10–30 s	Saliva	[38]
2	ITO	Antibody	Au NPs@rPGO	100 nmol/L–500 fmol/L	39.5 fmol/L	nr ^3^	Serum	[39]
3	CSPE	Protein	Ni(OH)_2_ NPs	1 fg/mL to 1 µg/mL	0.3 fg/mL	20 min	Serum	[40]
4	SPE	Clinical samples	Au NS & GO	0–1800 × 10^−20^ μg/mL	1.68 × 10^−22^ μg/mL	1 min	Blood, saliva, and oropharyngeal/nasopharyngeal swab	[41]
5	Au-SPCE	ssDNA	BNQDs/FGNs	10^−18^–10^−9^ M	0.48 aM	30 min	Nasopharyngeal swab	[42]
6	CSPE	Dithiolated DNA and RNA	AuNTs	0–500 fM	22.2 fM	1.5 h	Nasopharyngeal swab	[43]
7	Laser-scribed graphene (LSG)	Antibody	AuNS	5.0–500 ng/mL	2.9 ng/mL	nr	Serum	[44]
8	SPE	Aptamer	SWCNT	0.3–300 nM	7 nM	nr	Nasopharyngeal swab	[45]
9	Glassy carbon electrode (GCE)	Whole virus of SARS-CoV-2	GO	0.1933–2.708 µg /mL	0.1802 fg/mL	1 min	Plasma blood	[46]
10	CSPE	Antibody	MBs	0.01–0.6 μg/mL	19 ng/mL	30 min	Saliva	[47]

^1^ Linear dynamic range (LDR). ^2^ Limit of detection (LOD). ^3^ nr—not reported.

**Table 4 micromachines-14-00933-t004:** Representative examples of recent CA for the detection of SARS-CoV-2.

SI. No.	Electrode Platform	Bioreceptors	Nanomaterials	Characteristics				Ref.
				LDR ^1^	LOD ^2^	Time Detection	Clinical Samples	
1	Sensor-modified electrode chip	ssDNA	GNPs and AuNPs	585.4 copies/μL to 5.854 × 10^7^ copies/μL	6.9 copies/μL	<5 min	Nasal swab and Saliva	[50]
2	Screen-printed on a PET substrate	Antibody	nr ^3^	600 pg/mL–60 μg/mL for IgG and 500 pg/mL –50 μg/mL for IgM	10.1 ng/mL for IgG and 1.64 ng/mL for IgM	13 min	Serum	[51]
3	SPE	Antibody	nr	0.15 to 100 ng/mL	0.15 ng/ mL	nr	Serum	[52]
4	Au-SPE	Antibody	Magnetic nanobeads	1 ng/mL– 10 ng/mL	50 pg/mL	<1 h	Serum	[53]
5	Rotating disk electrode (RDE)	Protein	NiOOH	0.74–0.074 fg/mL	0.074 fg/mL	100 milliseconds	Saliva	[54]

^1^ Linear dynamic range (LDR). ^2^ Limit of detection (LOD). ^3^ nr—not reported.

## Data Availability

Data sharing is not applicable to this article as no new data were created or analyzed in this study.

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
