# Peer review of "Using Nanomaterials for SARS-CoV-2 Sensing via Electrochemical Techniques"

_micromachines, 2023, doi:10.3390/mi14050933_

Round 1
Reviewer 1 Report
The present review summarized the recent advances on electrochemical techniques for SARS-CoV-2 diagnosis. The literature cited are extensive and representative. The figures are finely reproduced and the figures are precise and informative. The perspectives are logical and enlightening. With fluent writing, the manuscript can be accept after some minor revisions.
1\“In vitro “and “in vivo” should use italic forms.
2\ Mark “A” was missed in the small image at the upper left corner of Figure 2.
3\ Line 120: use “EIS” instead of “Electrochemical impedance spectroscopy(EIS)”, since it is not the first appearance.
4\Line 144: use Carbon SPE(CSPE) .
5\ For table2, it is suggested that the studies should be classify according to the type of test sample or electrode,and list with better logical orders.
Author Response
"Please see the attachment."

Reviewer 2 Report
The authors report a short but meaningful review regarding electrochemical detection methods for SAR-CoV-2. The review provides fundamental knowledge of electrochemical techniques for future applications. More than that the authors provide a comprehensive introduction of biosensors and electrochemical biosensors as well a thorough presentation of SAR-CoV-2. Furthermore, the authors make a structured description of Electrochemical biosensing with explanative figures and well-structured tables. Nonetheless, the 3rd section “The importance of nanomaterials for developing biosensors for SARS-CoV-2” is poor described and papers of actuality in this field are not mentioned. For examples, th e authors mention graphene but fails to mention the studies conducted by Zhang et al. which reported the fabrications of an immunosensor using the combination of graphene-FET with antigen–antibody interaction that can give real-time detection by interacting with SARS-CoV-2 spike protein (doi.org/10.48550/arXiv.2003.12529 ) or by other authors such as Li et al (10.1016/j.bios.2021.113206) which reported the development of a Rapid and unamplified identification of COVID-19 with morpholino-modified graphene field-effect transistor nanosensor. The same lack of attention was awarded to the other nanomaterials as well.
Even though, overall the review is well structured and reunites meaningful information to the field, the authors should consider adding some more information to the 3rd and 4th sections.
Please see and refer to:
doi: 10.1007/s41664-021-00200-0;
doi: 10.1016/j.scitotenv.2020.142363;
doi.org/10.3390/bios12080637;
doi.org/10.1002/EXP.20210232;
DOI: 10.1039/D2RA01293F
etc.
There are plenty examples that could be used in the present manuscript.
Author Response
"Please see the attachment."

Round 2
Reviewer 2 Report
The authors have responded properly to all my comments.